# H3.3K27M-induced chromatin changes drive ectopic replication through misregulation of the JNK pathway in *C. elegans*

Kamila Delaney [1], Maude Strobino [1], Joanna M. Wenda [1], Andrzej Pankowski [2] & Florian A. Steiner [1]

Substitution of lysine 27 with methionine in histone H3.3 is a recently discovered driver mutation of pediatric high-grade gliomas. Mutant cells show decreased levels and altered distribution of H3K27 trimethylation (H3K27me3). How these chromatin changes are established genome-wide and lead to tumorigenesis remains unclear. Here we show that H3.3K27M-mediated alterations in H3K27me3 distribution result in ectopic DNA replication and cell cycle progression of germ cells in *Caenorhabditis elegans*. By genetically inducing changes in the H3.3 distribution, we demonstrate that both H3.3K27M and pre-existing H3K27me3 act locally and antagonistically on Polycomb Repressive Complex 2 (PRC2) in a concentration-dependent manner. The heterochromatin changes result in extensive gene misregulation, and genetic screening identified upregulation of JNK as an underlying cause of the germcell aberrations. Moreover, JNK inhibition suppresses the replicative fate in human tumor-derived H3.3K27M cells, thus establishing *C. elegans* as a powerful model for the identification of potential drug targets for treatment of H3.3K27M tumors.

[1] Department of Molecular Biology and Institute for Genetics and Genomics in Geneva, University of Geneva, 1211 Geneva, Switzerland. [2] Team of Mathematics and Physics, Faculty of Civil Engineering, Mechanics and Petrochemistry, Warsaw University of Technology, 09-400 Płock, Poland. Correspondence and requests for materials should be addressed to F.A.S. (email: Florian.Steiner@unige.ch)

Establishing specific chromatin landscapes to regulate access to the genetic material is an effective and dynamic mechanism to control cell fate and preserve cell identity. Nucleosomes are the basic structural and functional units of this chromatin regulation. They consist of an octameric core of histone proteins that provide a structural scaffold for the organization of DNA. In addition, histones can carry epigenetic information, either through post-translational modifications (PTMs) of the N-terminal tails or through the incorporation of histone variants. These chromatin signatures are indispensable for many crucial cellular processes like DNA replication, transcription, cell division or differentiation[1,2]. Histone H3.3 is a major variant of histone H3 that is mainly associated with regions of high nucleosome turnover[3], and loss of H3.3 results in severe sterility or lethality phenotypes in most organisms[4,5]. H3.3 is highly conserved in plants, animals and fungi and is distinguished from canonical H3 by only a few key amino acids that are important for the association with the H3.3-specific histone chaperones HIRA and DAXX[6–8].

Despite the importance for chromatin biology, only a few mutations in histone genes are directly associated with specific diseases. One notable example is the recently discovered replacement of lysine 27 with methionine in histone H3 or, more commonly, H3.3 that acts as a driver mutation of specific types of pediatric diffuse intrinsic pontine gliomas (DIPGs)[9] and some cases of acute myeloid leukemia[10]. In tumor cells carrying this oncohistone, trimethylation of lysine 27 on histone H3 (H3K27me3) is strongly and globally depleted from chromatin[11–13].

H3K27me3 is a mark characteristic for facultative heterochromatin, and is associated with transcriptionally repressed regions[14]. H3K27 methylation is deposited, recognized and propagated by the Polycomb repressive complex 2 (PRC2), a multiprotein complex responsible for maintaining the silent state of the genes during development and cell differentiation[15,16]. PRC2-mediated H3K27me3 propagation is the result of dynamic interactions between the PRC2 complex and pre-existing H3K27me3, which allosterically activates PRC2 and facilitates the spreading of the mark to neighboring nucleosomes[17–21].

The observed depletion of H3K27me3 in H3.3K27M tumor cells was initially explained by an increased affinity of PRC2 to the K27M-containing H3.3, resulting in PRC2 trapping on the nucleosomes containing this oncohistone[11,13,22]. Sequestration of PRC2 can explain how H3.3K27M, present only in a small fraction of all nucleosomes, acts as a dominant negative factor to affect H3K27me3 levels genome-wide. However, retention of the mark in some genomic regions of mutant cells is inconsistent with the hypothesis that PRC2 is immobilized by H3.3K27M-containing nucleosomes, and instead suggests that part of the PRC2 pool remains active to maintain H3K27me3 levels at some PRC2 targets[11,12]. Moreover, ChIP-seq analysis showed that PRC2 is excluded from, rather than immobilized on H3.3K27M-containing nucleosomes, and residual PRC2 activity is one of the factors promoting tumorigenesis[23]. Regardless of the exact mechanism of PRC2 inhibition, several studies revealed H3.3K27M-induced changes of H3K27me3 levels at gene promoters and enhancers, showing that active promoters, which are enriched in H3.3 incorporation and H3K27ac, tend to lose H3K27me3 while poised enhancers tend to gain the mark[11,23,24]. Moreover, these studies have led to a detailed understanding of how H3K27 methylation states affect gene expression in tumor cells. However, several key aspects of how H3.3K27M mutation leads to the formation of tumors are still not understood[25]. First, a precise understanding of how oncohistone incorporation remodels the H3K27me3 landscape genome-wide is lacking. Second, a causal link between the chromatin changes and the cancerous cell fate at a transcriptional level has not been established. And third,

the cell-type specificity of H3.3K27M-driven tumorigenesis still remains to be elucidated. The available model systems to study consequences of the H3.3K27M mutation either do not result in a replicative fate or require additional mutations (e.g., mutation of p53 and PDGFRA) to induce tumorigenesis[26–29].

To overcome these limitations and to investigate these key outstanding questions, we introduced the H3.3K27M mutation into *C. elegans*. Mutant worms display ectopic DNA replication and cell-cycle progression in the germline. We thus provide an animal model with a tumor-like phenotype induced by the oncohistone that does not require secondary mutations. Using this model system, we establish that PRC2 activity in mutant cells is affected both by H3.3K27M incorporation and the pre-existing levels of H3K27me3 in a concentration-dependent manner. Moreover, we identify the JNK pathway as an important link between the global reorganization of heterochromatin and the replicative cell fate in both nematodes and human tumor-derived cell lines.

## Results

**H3.3K27M drives germcell aberrations and sterility in *C. elegans*.** We introduced the K27M mutation into the H3.3 gene *his-72*, one of five *C. elegans* H3.3 genes, which is ubiquitously expressed and non-essential[30]. *His-72/H3.3* transcript levels are 50 times lower than canonical H3 transcript levels, implying that only a fraction of all nucleosomes incorporates this H3.3 protein[31]. Worms carrying the H3.3K27M mutation show normal somatic development, but display almost fully penetrant sterility at 25 °C, indicative of a germ-line defect (Fig. 1a). The mutant worms that do not show complete sterility have strongly reduced brood sizes. The mutation is semidominant, as sterility is also observed in heterozygous animals and can be induced by delivering extrachromosomal copies of H3.3K27M (Supplementary Fig. 1). In wild-type *C. elegans* germ lines, germ cells derive from a distal stem cell, undergo a few cycles of replication and mitotic division and then mature through meiotic phases in an assembly line fashion into oocytes that are arrested in diakinesis of meiosis I until fertilization (Fig. 1b, left panel). DNA replication is normally completely absent in proximal germ cells and only resumes during embryogenesis. Remarkably, in the H3.3K27M mutant, germ lines develop without defects, but adult proximal meiotic germ cells adopt an ectopic replicative fate, causing endomitosis and sterility (Fig. 1b, right panel). Mutant germ cells first show abnormal appearance at the transition from pachytene to diakinesis of meiosis I. Mutant proximal germ lines contain an increased number of oocytes that accumulate DNA contents many-fold higher than wild-type oocytes (Fig. 1c, d). The presence of these endomitotic oocytes suggested an ectopic activation of DNA replication in mutant germ lines. Immunofluorescence experiments revealed an ectopic expression of DNA polymerase delta subunit 2 (POLD2) at late pachytene stage and in endomitotic oocytes (Fig. 1e). Ongoing replication was also evident from BrdU incorporation (Fig. 1e). Some, but not all oocytes with over-replicated genomes are positive for the mitosis marker histone H3 phosphoS10, indicative of aberrant cell-cycle progression (Fig. 1e). However, mitosis does not progress, and continuous replication results in DNA accumulation. We also detected a high number of foci containing the DNA-repair protein RAD-51 in endomitotic oocytes, indicating that the ectopic DNA replication results in extensive DNA damage (Fig. 1e). To investigate the DNA accumulation in more detail, we sequenced the genomic DNA of endoreduplicated and wild-type proximal gonads. We found no evidence for preferential replication of specific regions, indicating that the entire genome is evenly replicated (Supplementary Fig. 2). Taken together, ectopic activation of DNA replication, accumulation of DNA damage, and aberrant cell-cycle progression

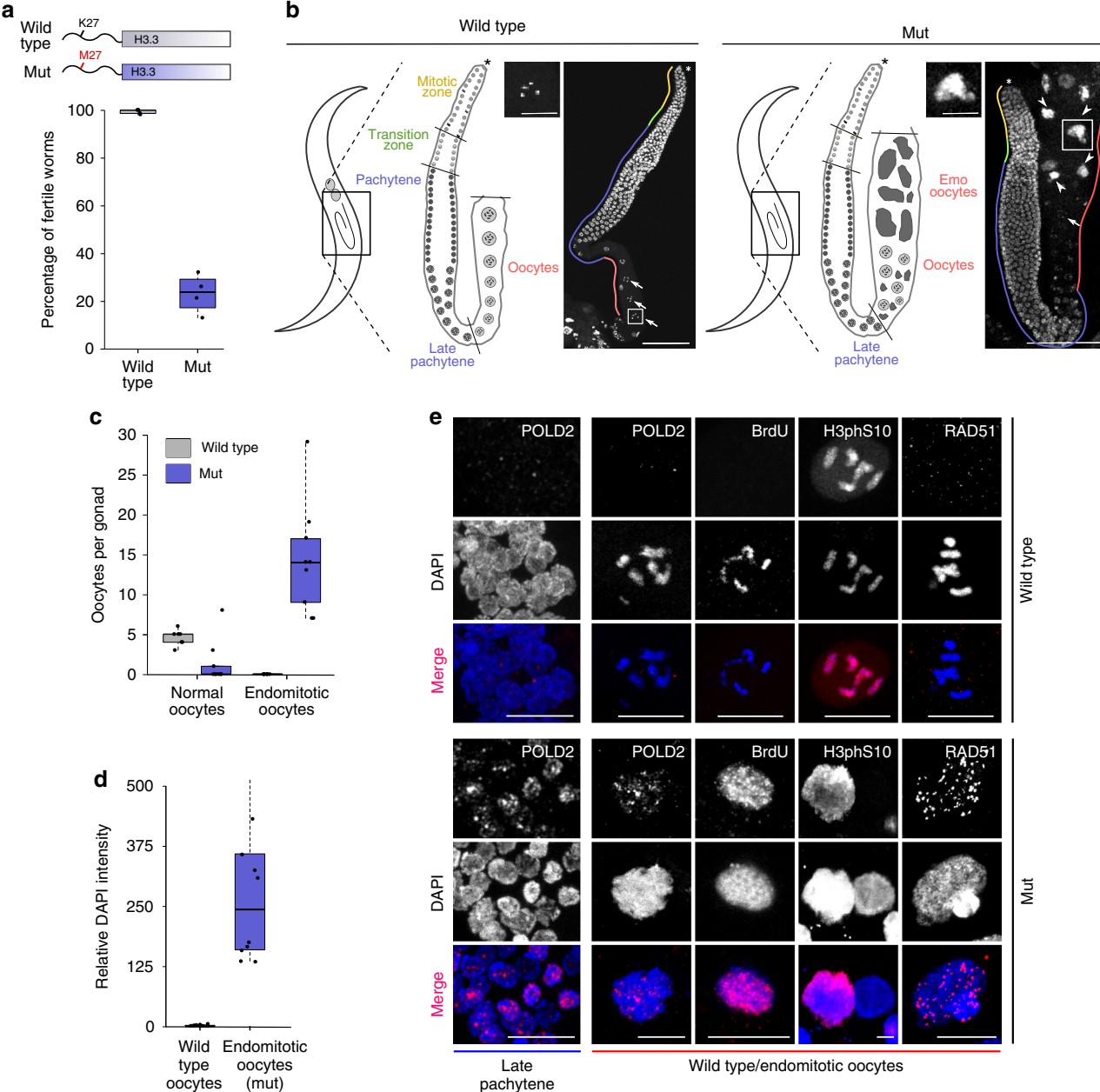

**Fig. 1** H3.3K27M mutation drives *C. elegans* germ cells towards a replicative fate. **a** Cartoon of H3.3K27M mutation, and boxplot showing fertility levels of wild-type and H3.3K27M mutant (mut) worms at 25 °C. $N = 5$ for wt, $N = 4$ for mut. **b** Cartoon images of worms and germ lines highlighting different stages of germcell development in wild-type and mut worms. DAPI-stained adult gonads of wild-type (left) and mut (right) worms. Distal tip (asterisk), normal (arrows), and endomitotic (arrowheads) oocytes are highlighted. A representative normal (wild type) and endomitotic (mut) oocyte is enlarged. Scale bars represent 75 μm on whole gonad pictures and 20 μm on enlarged oocyte pictures. **c** Boxplot showing the number of normal and endomitotic oocytes per gonad in wild-type and mut worms. $N = 7$ for wt, $N = 9$ for mut. **d** Boxplot showing relative DNA content in endomitotic oocytes compared with wild-type oocytes (diakinesis I, 4 n) based on the DAPI quantification. $N = 10$. **e** H3.3K27M mutant late pachytene and diakinesis germ cells are positive for markers of replication (DNA polymerase delta subunit 2 (POLD2) and BrdU incorporation), mitosis (H3 phosphoS10) and DNA double stranded breaks (RAD-51). POLD2 staining in pachytene nuclei as well as POLD2, BrdU, H3 phosphoS10, and RAD-51 staining in normal oocytes of wild-type and endomitotic oocytes of mut worms are shown. Scale bars equal 15 μm for late pachytene nuclei images and 10 μm for wild-type and endomitotic oocytes images

observed in H3.3K27M mutant worms recapitulate tumor-like characteristics, indicating that the H3.3K27M mutation alone can be sufficient to induce aberrant cell fates.

**Oncohistone incorporation regulates PRC2 activity and distribution.** To understand the cause of the observed germline phenotypes, we aimed to establish the consequences of

the H3.3K27M mutation for the H3K27me3 landscape before endoreduplication sets in. To unambiguously distinguish the wild-type and mutant versions of H3.3 from H3 in immunofluorescence and genomics experiments, we added an OLLAS epitope tag to HIS-72/H3.3. It has been previously shown that H3.3 is not evenly incorporated into the genome, but mainly associates with regions of open chromatin[32,33]. In *C. elegans* germ

cells, it is depleted from chromosome X[30,34]. This depletion is likely caused by the transcriptional repression of chromosome X, which is mediated mainly by the PRC2 complex through extensive H3K27 trimethylation[35,36]. MES-2, the worm homolog of the PRC2 subunit EZH2, normally shows a diffuse distribution in germ-cell nuclei, but strikingly, introduction of the H3.3K27M mutation causes an altered distribution and accumulation in distinct regions of the nuclei (Fig. 2a). The change in PRC2 localization is accompanied by a dramatic reorganization of H3K27me3, which becomes depleted from most of the chromatin, but remains enriched on chromosome X (identified by co-staining with H3K4me3) (Fig. 2a; Supplementary Fig. 3). This suggests that PRC2 is inhibited locally on the autosomes by the oncohistone incorporation, but that sufficient free PRC2 remains to maintain H3K27me3 on the chromosome X, where the H3.3 levels are low. These results also imply that oncohistone incorporation is the main regulator of PRC2 activity in mutant cells, and that only regions with little or no oncohistone incorporation retain H3K27me3.

**PRC2 activity changes upon altered oncohistone incorporation.** We next aimed to examine if PRC2 activity can be further reduced by altering the oncohistone incorporation genome-wide. It has previously been shown that the chaperone specificity for

H3 and H3.3 is conveyed by a four amino acid motif that differs between these two histone proteins, SAVM and AAIG, respectively[6–8,37]. We recently demonstrated that exchanging the H3.3-specific motif with the one present in H3 results in a more even distribution of the mutated H3.3 on the chromatin in *C. elegans* germ cells, and the H3.3-characteristic depletion from the X chromosome is no longer evident[30]. Introducing the K27M mutation into this H3-like histone therefore allowed us to test the effects of oncohistone redistribution without changing its levels (Fig. 2b). In the context of this H3-like oncohistone, PRC2/MES-2 remains concentrated at one region of the germ-cell nuclei. Immunofluorescence experiments showed that the H3-like oncohistone is indeed not depleted from the X chromosome (Fig. 2b). However, the altered oncohistone incorporation pattern does not seem to further disrupt the H3K27me3 landscape, and H3K27me3 levels remain high on the X chromosome (Fig. 2b). Consistently, total H3K27me3 levels are strongly reduced in both H3.3K27M and H3-like K27M mutants compared with wild type, while wild-type and mutant versions of H3.3 are present at similar levels (Fig. 2c).

Altering the oncohistone incorporation from an H3.3 to an H3-like distribution resulted in a superficially unchanged H3K27me3 distribution, and fertility levels remained at about 20% compared with wild type (Fig. 2d). However, oocyte

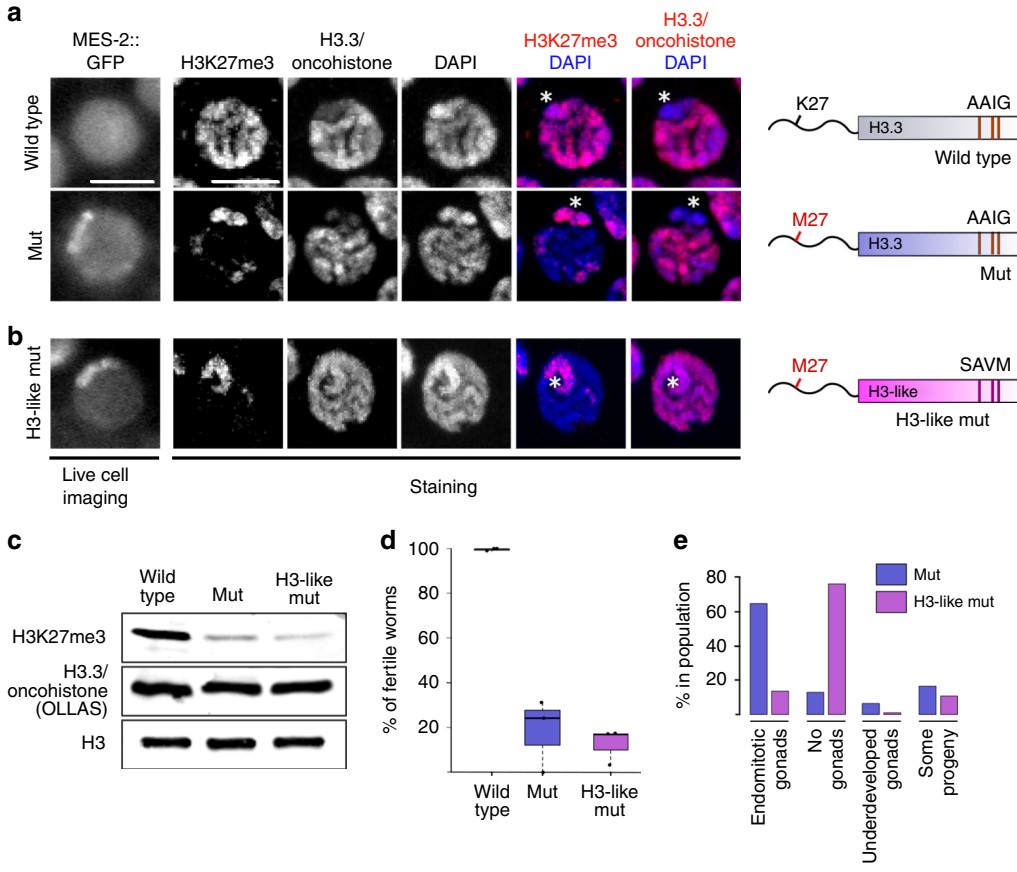

**Fig. 2** Oncohistone incorporation patterns induce changes in nuclear PRC2 distribution and sterility phenotypes. **a, b** Live cell imaging of GFP-tagged MES-2/EZH2 (the catalytic subunit of *C. elegans* PRC2), and immunofluorescence of H3K27me3 and H3.3/oncohistone in pachytene nuclei of wild-type and H3.3K27M mutant (mut) (**a**) and H3-like K27M oncohistone (H3-like mut) (**b**) worms. Scale bars represent 5 μm. Chromosome X was identified by depletion of H3.3 and H3K4me3 staining shown in Supplementary Fig. 3, and is marked with an asterisk. The cartoons on the right illustrate the mutations introduced into the H3.3 protein in each strain. **c** Western blot of H3K27me3 and H3.3/oncohistone levels in wild-type, mut, and H3-like mut worms. All versions of H3.3 are OLLAS-tagged to distinguish them unambiguously form H3. H3 levels are shown as loading control. **d** Boxplots illustrating levels of fertility of wild-type (gray), mut (blue), and H3-like mut (purple) worms at 25 °C. N = 3 for all conditions. **e** Bar plots showing types of sterility observed in mut (blue) and H3-like mut (purple) worms at 25 °C

endomitosis is not the prevalent cause of sterility in H3-like K27M mutant worms. Instead, a high percentage of the animals fail to develop a gonad, similar to what is observed in PRC2 null mutants (Fig. 2e)[38]. This result implies that the different incorporation patterns of the H3.3 and H3-like oncohistone cause different cell fates: ectopic replication in case of the former and non-proliferation in case of the latter, indicating that not only oncohistone levels, but also oncohistone distribution are important for the phenotypes observed in K27M mutant cells.

**Oncohistone and H3K27me3 act antagonistically on PRC2 *in cis*.** Despite causing strikingly different germ-cell fates, introduction of the K27M mutation in either H3.3 or H3-like histone results in superficially similar changes in H3K27me3 distribution, with a depletion from the autosomes and maintenance of high levels on chromosome X. To investigate these changes in more detail on a genomic scale, we performed chromatin immunoprecipitation followed by high-throughput sequencing (ChIP-seq), probing for H3K27me3 enrichment and oncohistone occupancy. As previously reported for mammalian oncohistones, the K27M mutation alone does not alter the H3.3 distribution pattern (Fig. 3a, Supplementary Fig. 4)[13]. H3.3 is enriched on autosomes and depleted from the X chromosome (Fig. 3a), consistent with the finding that H3.3 occupancy levels correlate with gene expression, and that autosomes have higher average gene expression levels[21,39]. Previous work showed that H3K27me3 domains are anti-correlated with domains of H3K36me3, and that the H3K36me3 domains set boundaries for the spreading of H3K27me3[36,40]. We therefore also analyzed the H3K36me3 patterns in wild-type and H3.3K27M mutant worms, and found that the genomic distribution and average occupancy levels remained unchanged (Supplementary Fig. 5). The observed phenotypes are therefore unlikely to be linked to changes in H3K36me3 distribution.

Upon changing the chaperone recognition motif from AAIG to SAVM, the resulting H3-like oncohistone shows an incorporation pattern with domains similar to H3.3, but these domains widen and spread into regions not normally occupied by H3.3 (Fig. 3a). Importantly, changing the chaperone recognition motif by itself does not affect H3K27me3 patterns (Supplementary Fig. 6). The genomic H3K27me3 analysis of the oncohistone mutant strains confirmed the immunofluorescence findings that the strongest depletion of the mark is observed on the autosomes, while the X chromosome remains strongly enriched for the mark in both the H3.3K27M and H3-like K27M mutants (Fig. 3a, b, Supplementary Fig. 4a). To visualize the dependencies between oncohistone incorporation and H3K27me3 levels, we divided the genome into four categories based on changes induced by the H3.3K27M mutation: domains that retained (i) or lost (ii) H3K27me3 signal upon oncohistone incorporation, domains with no H3K27me3 occupancy in neither wild-type nor mutant worms (iii), and the X chromosome (iv) (Fig. 3a, c).

In H3.3K27M mutant worms, H3K27me3 appears inversely correlated with oncohistone incorporation. H3K27me3 occupancy is low or becomes lost in domains with high oncohistone incorporation (domains ii and iii), and remains high in domains with low oncohistone incorporation (domains i and iv) (Fig. 3a, c). These observations support a model of local, dose-dependent PRC2 inhibition in regions with oncohistone incorporation[28]. To test this model, we also analyzed the H3K27me3 levels in the H3-like K27M mutant, where oncohistone incorporation is increased in the domains that normally don't contain H3.3 and on the X chromosome (domains i and iv) (Fig. 3a, c). Surprisingly, this increase only affects H3K27me3 levels on autosomes, not on the

X chromosome (Fig. 3a, c; Supplementary Fig. 4a), suggesting that oncohistone incorporation is not the only factor affecting PRC2 activity, but that high pre-existing concentrations of H3K27me3 can stimulate the enzyme sufficiently to maintain its activity.

To test whether H3K27me3 in the mutants indeed depends on both oncohistone incorporation and pre-existing (wild type) H3K27me3 levels, we performed hierarchical clustering of the ChIP-seq signal in 10 kb bins covering the entire genome. For both oncohistone mutants, the clustering was performed on three datasets: wild-type (pre-oncohistone mutation) H3K27me3 levels, oncohistone levels, and H3K27me3 levels upon oncohistone incorporation. Analysis of the H3.3K27M mutant data confirmed that oncohistone incorporation and PRC2 activity are largely antagonistic (Fig. 3d, left panel). On autosomes, positions with high levels of oncohistone incorporation show very low levels of H3K27me3 (Fig. 3d, left panel, clusters 1–3). Positions with low oncohistone incorporation on both autosomes and chromosome X, however, retain high levels of H3K27me3 (Fig. 3d, left panel, clusters 4–5). Interestingly, a small cluster on the X chromosome retains H3K27me3 despite high levels of oncohistone incorporation (Fig. 3d, left panel, cluster 6), suggesting that high H3K27me3 levels can overcome the inhibitory effect of the oncohistone incorporation. This observation is consistent with the local stimulatory role of H3K27me3 for PRC2 activity described recently[18,20].

In the H3-like K27M mutant, oncohistone incorporation appears more evenly distributed across the genome (Fig. 3d, right panel). This changed incorporation pattern results in further local reduction of the H3K27me3 on the autosomes, reflected by enlargement of cluster 3 of the heat map (Fig. 3d, right panel). However, some autosomal regions retain high H3K27me3 levels despite increased oncohistone occupancy (Fig. 3d, right panel, cluster 4). Similarly, the high H3K27me3 signal persists throughout the X chromosome, despite the increased local oncohistone levels (Fig. 3d, right panel, clusters 5–6). These results confirm that high levels of pre-existing H3K27me3 can locally sufficiently stimulate PRC2 to overcome the inhibitory effects of the oncohistone incorporation. Taken together, we find that PRC2 activity is influenced locally by both initial levels of H3K27me3 and oncohistone incorporation (Fig. 3e). These factors are sufficient to explain the oncohistone-induced H3K27me3 changes genome-wide and provide a mechanistic explanation for the aberrant H3K27me3 patterns observed in mutant worm and vertebrate cells.

The hierarchical clustering method allowed us to dissect the dependencies between the H3K27 methylation states and oncohistone incorporation patterns, but it did not quantitatively establish that oncohistone-induced changes follow the same pattern in both mutants. Moreover, we were curious to see whether the relationship between these factors of PRC2 regulation also hold true in a mammalian system. To address these questions, we utilized a Taylor's theorem-based approximation and modeled the non-linear relationship between oncohistone incorporation and H3K27me3. The model was developed on the *C. elegans* H3.3K27M mutant ChIP-seq results and then also applied to the datasets obtained for the H3-like mutant and recently published results from H3.3K27M-containing mouse cell lines[24]. Although the distribution of the oncohistone incorporation is different in each case, the function approximates the observed H3K27me3 distribution for all three datasets in a similar way (Supplementary Fig. 7). This confirms that, in all cases, the inhibitory effects of oncohistone incorporation are dominant in regions with low pre-existing H3K27me3, but become negligible in regions of high pre-existing H3K27me3, regardless of the oncohistone distribution.

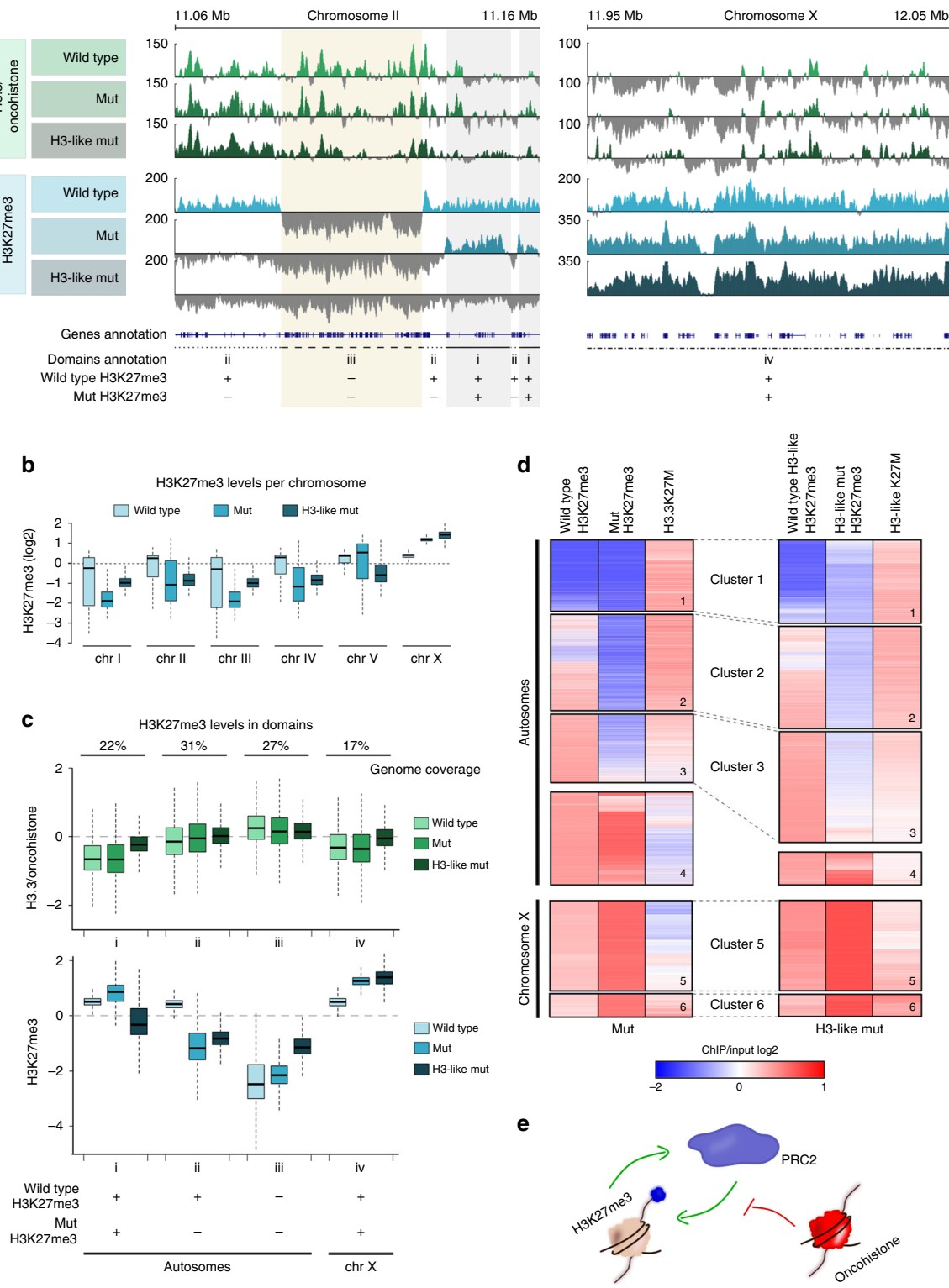

**H3.3K27M-induced phenotypes are driven by JNK misregulation**. H3.3K27M-mediated H3K27me3 redistribution leads to extensive gene misregulation in human cells[9,41]. Many of these misregulated genes are implicated in processes potentially driving tumorigenesis, such as mitogen-activated protein (MAP) kinase signaling or cell cycle control, and misregulation of some of them, like tumor suppressor Wilms Tumor 1, were shown to promote proliferation of the tumor-derived cells[24]. However, a clear understanding of how changes in gene expression result in tumorigenesis is still lacking, especially since many of the animal models require secondary mutation for the activation of cell cycle programs. The *C. elegans* model presented here contains the H3.3K27M mutation in an otherwise wild-type background, thus potentially allowing us to link the effect of the mutation to specific pathways that drive the cells towards their aberrant fate. We identified about 500 genes with changes in expression upon

**Fig. 3** Oncohistones and H3K27me3 antagonistically regulate PRC2 activity *in cis*. **a** Genome browser views of H3.3/oncohistone (green tracks, top panel) and H3K27me3 (blue tracks, bottom panel) occupancy in wild-type (light), H3.3K27M mutant (mut, medium), and H3-like K27M mutant (H3-like mut, dark) worms (100 kb windows). Genes and H3K27me3 domain annotations are highlighted at the bottom of the genome view. The domains are defined based on comparison of H3K27me3 occupancy between wild type: (i) autosomal signal is maintained, (ii) autosomal signal is lost, (iii) autosomal signal never present, (iv) chromosome X signal is maintained. **b** Boxplots showing H3K27me3 occupancy per chromosome in wild type (light blue), mut (medium blue), and H3-like mut (dark blue). **c** H3.3/oncohistone (green) and H3K27me3 (blue) occupancy in identified domains in wild type (light), mut (medium), and H3-like mut (dark). **d** Heat map showing hierarchical clustering of 10 kb bins covering the entire genome in mut (left panel) and H3-like mut (right panel) worms based on levels of H3K27me3 before and after acquiring oncohistone mutation, and oncohistone incorporation. Autosomes and X chromosome were clustered and are shown separately. **e** Model for oncohistone-mediated regulation of PRC2. PRC2 activity is locally both positively regulated by existing H3K27me3 and inhibited by incorporated oncohistones, and the *in cis* balance between these two factors determines the pattern of H3K27me3 in mutant cells

acquisition of the H3.3K27M mutation (Fig. 4a, Supplementary Data 1). Changes in gene expression are anti-correlated with changes in H3K27me3 occupancy (Fig. 4b, c), and upregulated genes are mainly located on the autosomes, while downregulated genes are enriched on chromosome X (Supplementary Fig. 8). Interestingly, H3K27me3 levels are gained or lost over the entire gene body (Fig. 4b). These results imply that the gene expression changes are likely a direct consequence of the H3K27me3 redistribution in the H3.3K27M mutant strain, consistent with the finding that misregulated genes are enriched for PRC2 targets in H3.3K27M tumor cells[41]. We separated all genes into expression quintiles based on expression levels in wild type, and plotted the average H3.3/oncohistone and H3K27me3 occupancies for each quintile (Fig. 4d, e). We confirmed that H3.3 levels are positively correlated with gene expression levels, and that H3.3 occupancy is unaffected by the K27M mutation (Fig. 3d, e; left panels). In contrast, H3K27me3 levels are lower in the H3.3K27M mutant worms compared with wild type (Fig. 3d, e; right panels). Given the interplay between the oncohistone and PRC2 described above, we expected genes with intermediate levels of both oncohistone and H3K27me3 occupancy to be most susceptible to chromatin changes. Indeed, almost 70% of the genes with changed expression in H3.3K27M mutant worms fall into gene expression quintiles II and III, where H3.3 and H3K27me3 levels are both present (Fig. 4f). Moreover, almost none of the misexpressed genes are germ-line specific[42], confirming that misregulation affects genes that are normally repressed in the germ line (Fig. 4g). The chromatin changes affect the expression of many specific gene groups and pathway components, but analysis of significantly enriched GO terms revealed that the most affected categories fall into kinase-related signal transduction proteins (Fig. 4h). However, from this data it is difficult to identify expression changes in specific genes that are causal for the observed phenotype.

To overcome this limitation and identify key genes involved in aberrant oocyte replication within these pathways in an unbiased way, we performed a random mutagenesis screen for genetic suppressors of H3.3K27M-induced sterility (Fig. 5a). We isolated several potential suppressor strains, but were able to map the causal mutation only for the allele with the strongest suppressor phenotype. The identified mutation is a serine 287 to asparagine substitution in KGB-1 (KGB-1 S287N) that acts as a potent suppressor of endomitosis that restored fertility in worms carrying the oncohistones (Fig. 5b). We confirmed this finding with an additional KGB-1 S287N allele generated by CRISPR/Cas9 mutagenesis that restored fertility to similar levels. KGB-1 is a homolog of mammalian Jun amino-terminal kinase (JNK), a stress-activated MAPK subfamily serine-threonine kinase[43]. In *C. elegans*, KGB-1 positively regulates the activator protein-1 (AP-1) transcription factor FOS-1[44], which in turn controls oocyte ovulation via IP3 signaling[45]. Misregulation of this regulatory cascade was shown to result in disruption of the $Ca^{2+}$

homeostasis in the germ line and in an oocyte endomitosis phenotype similar to the one observed in H3.3K27M worms[45].

To confirm that the H3.3K27M mutation leads to aberrant calcium signaling, we measured $Ca^{2+}$ levels using the indicator dye Calcium Green-1 dextran, injected into the worm gonads. We found that H3.3K27M mutant worms showed elevated levels of $Ca^{2+}$ in the germ line compared with wild type. This defect was partially rescued by the KGB-1S287N mutation, confirming that the replicative phenotype observed in these mutants correlates with aberrant $Ca^{2+}$ signaling, a known downstream effector of the JNK pathway (Fig. 5c)[45,46].

In H3.3K27M mutant germ lines, the *kgb-1* gene loses H3K27me3 signal, and *kgb-1* expression levels are significantly elevated (Fig. 5d, e). The KGB-1 S287N mutation does not restore the nuclear distribution of H3K27me3 observed in the oncohistone mutant (Fig. 5f, Supplementary Fig. 9), indicating that it rescues a defect downstream of the chromatin changes and may affect the stability or activity of the enzyme. Sequence alignment and comparison to the structure of mammalian JNK-1 indicated that the S287N suppressor mutation localizes in the serine/threonine kinase domain of the KGB-1, and is likely exposed on the surface of the protein, thus potentially serving as a phosphorylation target that regulates the activity of the enzyme (Supplementary Fig. 10)[47,48]. To test whether the sterility is driven by diminished activity of the JNK pathway, we reduced the expression levels of *kgb-1* by RNAi, which resulted in a partial rescue of the endomitosis phenotype and in a significant increase in fertility of H3.3K27M worms (Fig. 5g). To further validate the JNK pathway as potential target for antagonizing the effects of the H3.3K27M mutation, we treated worms with the JNK inhibitor SP600125[49–51]. Consistent with the RNAi experiments, the drug was able to significantly increase the number of fertile animals in the H3.3K27M population, while not affecting wild-type worms (Fig. 5h). Taken together, our results demonstrate that H3.3K27M-mediated redistribution of H3K27me3 directly results in gene expression changes, but that the aberrant oocyte replication and endomitosis is driven by misregulation of specific serine/threonine kinases such as JNK. We show that targeting this kinase by RNAi or by chemical inhibitors is an effective way to prevent the aberrant entry of oocytes into the cell cycle and restore fertility in H3.3K27M mutant worms.

**JNK inhibition is effective on human glioma-derived H3.3K27M cells.** Interestingly, JNK is one of the known targets for inhibiting proliferation of glioblastoma cells[49–51]. This suggests that in addition to the similarities in the H3.3K27M-induced chromatin changes between nematodes and vertebrates, similar downstream gene networks are misregulated to drive replicative cell fates. To explore this possibility, we treated three glioma cell lines, SF8628 derived from an H3.3K27M tumor, and SF9402 and SF9427 derived from tumors with wild-type H3.3, with the JNK

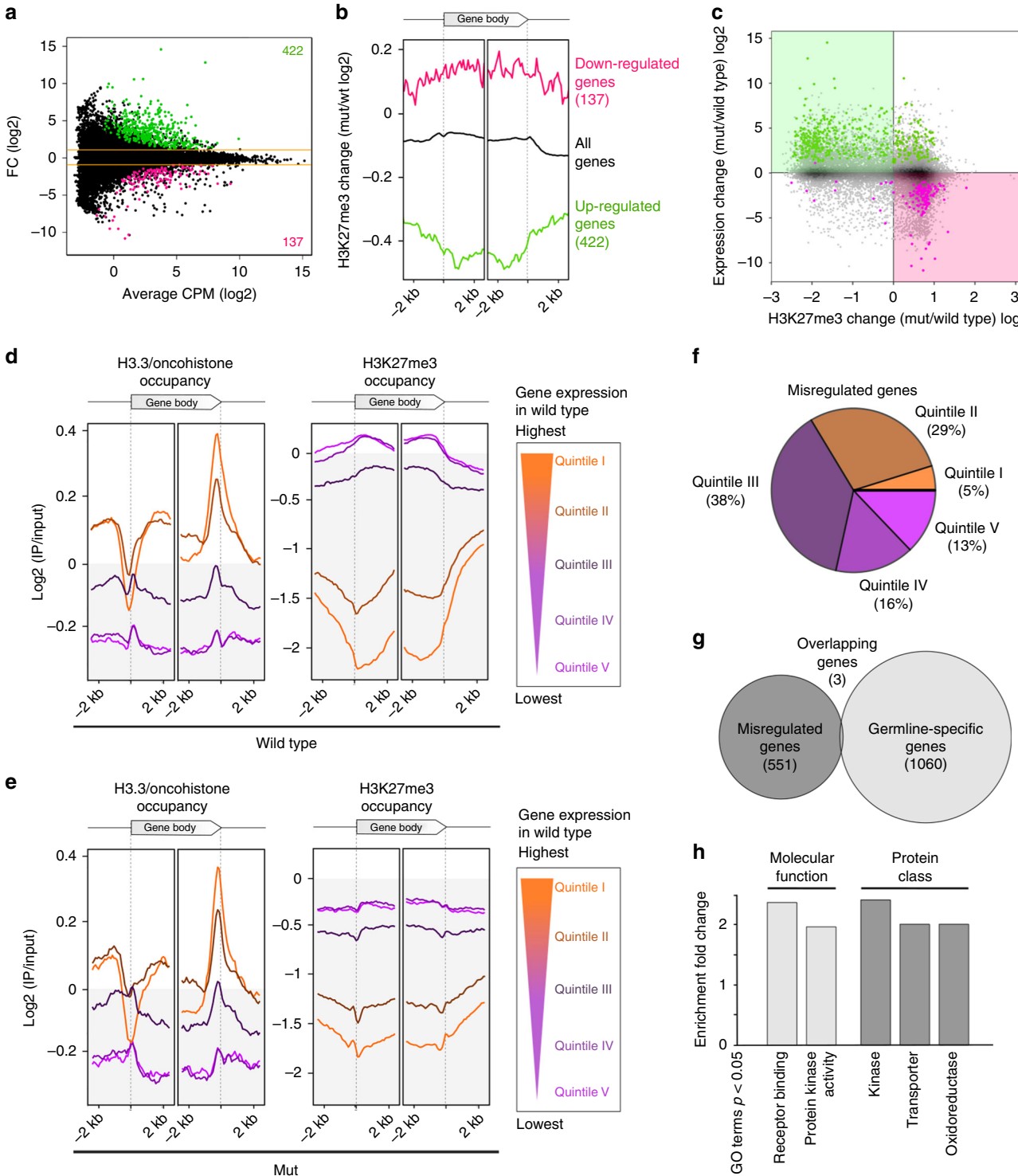

**Fig. 4** H3.3K27M mutation leads to extensive gene misregulation. **a** MA plot of RNA sequencing results showing genes up- (green) and downregulated (magenta) in H3.3K27M mutant (mut) compared with wild-type worms. **b** Average plots of H3K27me3 occupancy changes in mut compared with wild-type worms at all genes, and at genes up- (green) or downregulated (magenta) upon H3.3K27M mutation. **c** Scatter plot of H3K27me3 and expression changes upon H3.3K27M mutation for all genes. Upregulated genes are marked with magenta, and downregulated genes are marked in green. **d**–**e** Average plots showing H3.3 and H3K27me3 occupancy over gene bodies, with genes divided into quintiles based on wild-type levels of expression, in wild-type worms (**d**) and mut worms (**e**). **f** Pie chart showing proportions of misregulated genes in mut worms in each expression quintile. **g** Venn diagram showing the overlap between genes misregulated in mut worms and germ-line-specific genes[42]. **h** Bar plot showing GO Terms significantly enriched among genes misregulated in mut worms. GO Terms were called in two categories (molecular function and protein class), and only GO Terms enriched ≥ 2 fold are shown

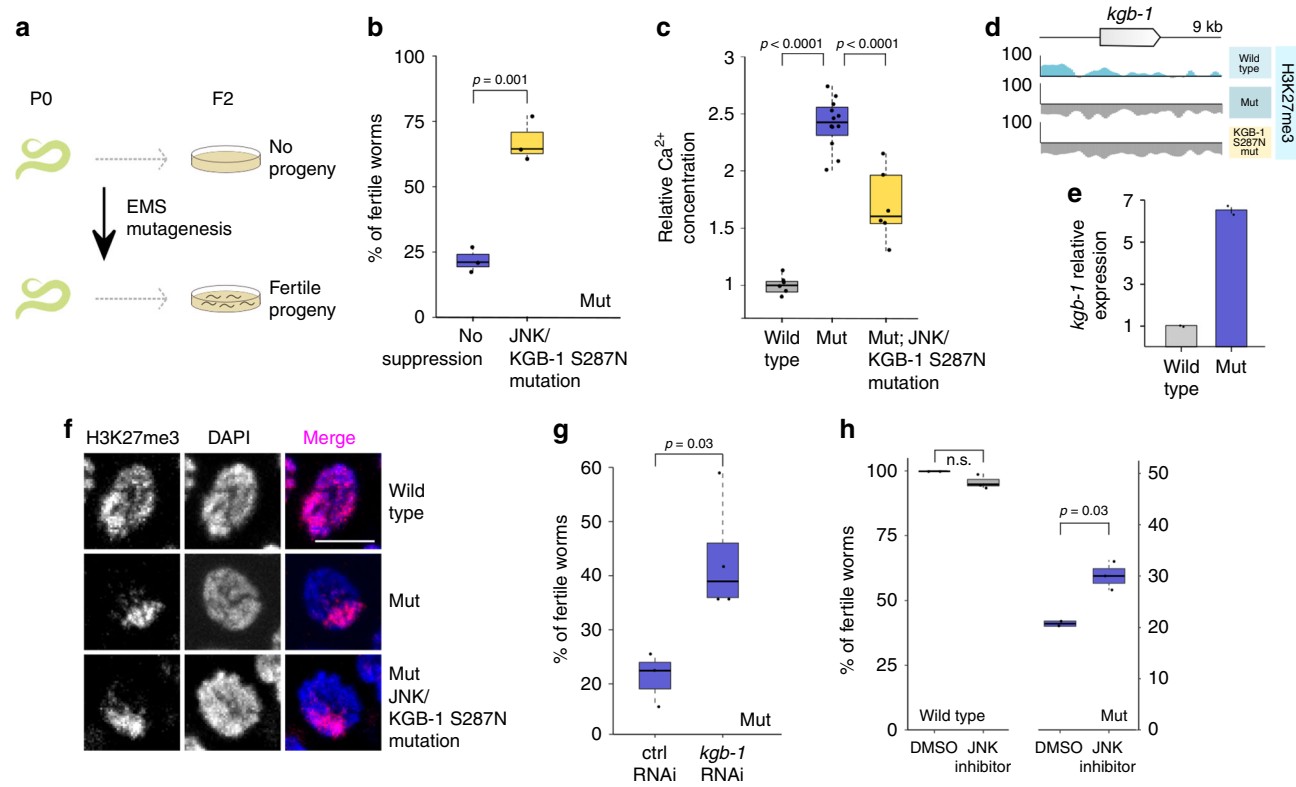

**Fig. 5** JNK inhibition counteracts phenotypes caused by H3.3K27M mutation. **a** Cartoon showing experimental setup for the random EMS mutagenesis. P0 worms were mutagenized, and individual F2 worms were screened for fertility that was maintained for several subsequent generations at 25 °C. **b** Fertility boxplot for H3.3K27M mutant (mut) worms with (yellow) and without (blue) KGB-1/JNK S287N suppressor mutation. $N = 3$ for all conditions. **c** Boxplot showing relative $Ca^{2+}$ concentration in gonads of wild type (gray), mut (blue) and mut; KGB-1/JNK S287N (yellow). Concentration in wild type was set as one and used as a reference for the other strains. $N = 6$ for wt, $N = 12$ for mut, $N = 6$ for mut; KGB-1/JNK S287N. **d** Genome browser view showing H3K27me3 levels around the *kgb-1* gene in wild-type, mut and mut; KGB-1/JNK S287N worms. **e** Bar plot showing the *kgb-1* expression levels in wild-type and mut germ lines based on RNA-seq (bottom). Error bars are standard deviations. **f** H3K27me3 staining in wild type, mut and mut; KGB-1/JNK S287N. Scale bar represents 5 μm. **g** Fertility boxplots of mut worms upon treatment with control (left) or *kgb-1*/JNK (right) RNAi. $N = 3$ for control, $N = 4$ for *kgb-1* RNAi. **h** Fertility boxplots of wild-type (gray) and mut (blue) worms upon treatment with DMSO or JNK inhibitor SP600125. $N = 3$ for inhibitor and $N = 2$ for DMSO treatment. Significance was tested for using student's *t*-test

inhibitor SP600125, using assays that were previously described to test differences between the same K27M-negative and -positive glioma cell lines[52]. As expected, pharmacological inhibition of the JNK pathway leads to a decrease in proliferation of non-H3.3K27M tumor cells, but remarkably, H3.3K27M cells are affected significantly more (Fig. 6). These results show that the anti-tumorigenic effects of JNK inhibition, established for glioblastoma, are particularly pronounced in cells harboring the H3.3 K27M mutation. These findings exemplify that the H3.3K27M nematode model can be utilized to identify relevant targets for future drug development through unbiased genetic screening.

## Discussion

Animal model systems are desirable to study the consequences of disease-associated mutations, as they provide a developmental context that cultured cells lack. The development of several H3.3K27M animal models, including mouse and *Drosophila*, have led to significant insight into H3.3K27M-mediated PRC2 inhibition. However, our understanding of the direct link between oncohistone incorporation, PRC2 inhibition and oncohistone-driven tumorigenesis remained incomplete due to inability of the oncohistone to induce a replicative phenotypes in these systems without introducing secondary mutations[26–29].

Introducing H3.3K27M into the nematode *C. elegans* resulted in global changes of the chromatin landscape and a striking replicative phenotype, allowing us to examine the relationships between oncohistone incorporation, loss of H3K27me3 and the aberrant cell fate. Interestingly, the replicative phenotype induced by the onco-histone is observed in the germ line rather than in neurons, as might have been expected from human tumors (Fig. 1). It has been previously observed that the induction of proliferation of post-mitotic cells in *C. elegans* is difficult to achieve[53]. *C. elegans* germ cells, however, appear relatively plastic, and their fate can be changed by modulating the activity of specific transcription factors and chromatin remodelers[54–56]. This germ-cell plasticity presumably also allows the observed ectopic DNA replication in our H3.3K27M system. Importantly, this phenotype does not require additional mutations, allowing us to directly attribute the observed changes to the presence of the oncohistone. The observed ectopic replication in the proximal part of the gonad differs from previously described tumorous germ-line phenotypes, where the distal, mitotic region of the gonad is extended[57]. The specific localization of the oncohistone-induced replicative fate is likely linked to the dereg-ulation of JNK signaling discussed below. The simple *C. elegans* model for the H3.3K27M mutation presented here allowed us to study the effect of oncohistone incorporation on the chromatin landscape and investigate the downstream effectors that promote the observed replicative germ-cell fate.

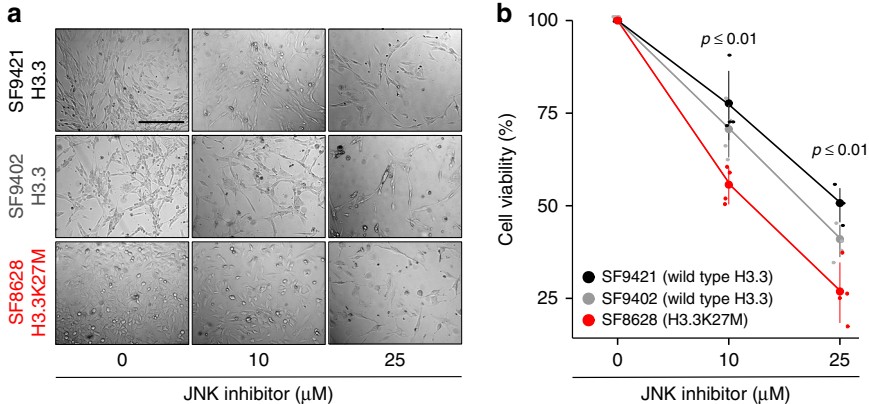

**Fig. 6** JNK inhibition decreases proliferation of human glioma-derived H3.3K27M cells. **a**, **b** Transmitted light pictures (**a**) and cell viability plots (**b**) of human glioma-derived cells with wild type (SF9421, black; SF9402, gray) or K27M mutant (SF8628, red) H3.3, treated with different concentrations of JNK inhibitor SP600125. $N = 4$ for all conditions. Significance was tested for using one-way ANOVA

Our analysis of H3.3K27M-induced chromatin changes allowed us to provide a general model for the concentration-dependent *in cis* interplay between pre-existing H3K27me3 and H3.3K27M incorporation that explains the genome-wide H3K27 methylation landscape in H3.3K27M mutant cells in both nematodes and vertebrates. We find that oncohistone incorporation antagonizes PRC2 and leads to a local loss of H3K27me3, but that domains of high H3K27me3 levels are maintained even upon oncohistone incorporation (Figs. 2, 3).

The maintenance of large domains of H3K27me3 and the accumulation of PRC2 on the X chromosome argue against the originally proposed model of physical trapping of PRC2 by H3.3K27M-containing nucleosomes[11,13,22]. Instead, our data suggest that the PRC2 inhibition is local and transient, and that substantial amounts of PRC2 remain active and are sufficient to maintain domains of high H3K27me3 or to locally increase H3K27me3 levels, as observed both in mammalian and worm systems. It has been recently shown that PRC2 appears to be excluded from K27M-containing nucleosomes[23,58], and that the presence of H3.3K27M histones prolongs the residence and search time of PRC2 on chromatin[59]. Structural studies revealed that PRC2 bridges neighboring nucleosomes. The substrate H3 tail is positioned near the catalytic subunit, while the already modified H3 tail of the neighboring nucleosome is positioned near the EED subunit and acts as an allosteric activator of the methyltransferase domain[20]. Our model predicts that in regions of high H3K27me3 occupancy and low oncohistone incorporation the allosteric activation of PRC2 activity and the spreading of H3K27me3 occurs without disruptions. However, with increasing local levels of the H3.3K27M, spreading of H3K27me3 becomes less efficient, resulting in decreased levels of H3K27me3 and decreased occupancy of PRC2. Free PRC2 can redistribute to regions with low oncohistone levels and H3K27me3-rich areas, where increased presence of the enzyme facilitates maintenance or local gains of H3K27me3 (Fig. 7). In support of our model, a recent study demonstrated that tethering PRC2 to regions that lost H3K27me3 upon H3.3K27M incorporation is able to locally restore methylation levels in these regions in mouse ESCs[24]. Taken together, the local, dynamic interplay between the effects of H3.3K27M and H3K27me3 on PRC2 activity determines the local concentration of PRC2 and influences whether or not H3K27me3 is propagated within a given region of the genome.

This model of PRC2 regulation by local levels of oncohistone and H3K27me3 appears simpler than the chromatin changes described for mouse and human cell culture models. This may

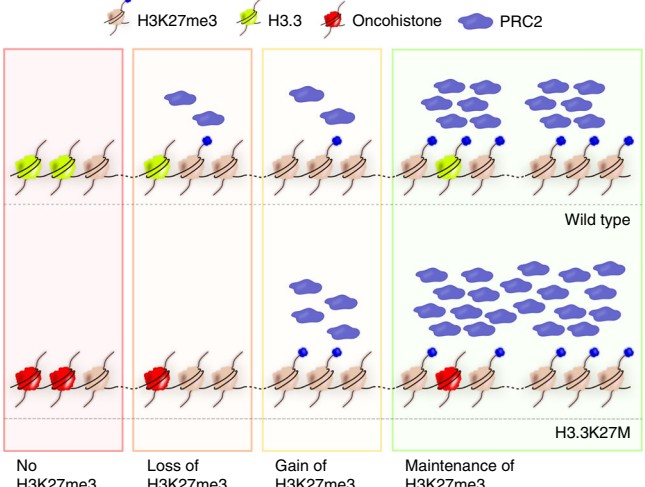

**Fig. 7** Model for local, concentration-dependent inhibition of PRC2 by H3.3K27M oncohistones. Incorporation of oncohistones locally disrupts the positive regulatory feedback loop between existing H3K27me3 and PRC2. Oncohistone presence in regions of low H3K27me3 results in a local reduction of H3K27me3 propagation by PRC2 and in a gradual loss of the mark. Free PRC2 can redistribute to regions with higher H3K27me3 occupancy, where it may result in local gains of the mark. In domains where levels of H3K7me3 are high, allosteric activation of PRC2 is sufficient to maintain the mark even in the presence of some oncohistone incorporation

be explained by the findings that DNA methylation, which is largely absent in *C. elegans*, also influences PRC2 activity in human cells[40]. H3K27me3 can also spread *in far-cis* via long-range contacts in human cells[60]. Such long-range interactions appear less common in *C. elegans* nuclei[61]. The simplicity of the spatial organization of the worm genome and the absence of DNA methylation therefore allowed us to establish the local dependencies between pre-existing methylation, oncohistone incorporation and PRC2 activity.

Somatic substitution of H3.3 lysine 27 with methionine can occur in every cell, yet this mutation causes tumorigenesis only in a few cell types and drives very specific types of brain tumors[25]. Our chromatin analysis showed that oncohistone-induced changes in H3K27 methylation are determined by two factors: the pattern of oncohistone incorporation and the pre-existing H3K27me3 states (Fig. 3). Both of these factors are linked to

transcriptional programs—H3.3 is correlated with active gene expression, and H3K27me3 is associated with transcriptionally repressed regions. Transcription profiles, and patterns of H3K27me3 and H3.3, greatly differ between cell types[62,63]. Therefore, the resulting H3K27me3 landscapes upon mutation of H3.3 will also differ, and it is tempting to speculate that these differences will result in different, cell type-dependent patterns of PRC2 inhibition, and cell type-specific phenotypes. These cell type-specific features may explain why H3.3K27M leads to ectopic activation of DNA replication and cell-cycle progression only in some neuronal cell types. The different outcome in cellular fate is exemplified in our worm model by the different germ-cell fates induced by H3.3 or H3-like pattern of the oncohistone incorporation (Fig. 2). Both induce the same spectrum of phenotypes, but the former results mostly in a replicative fate and endomitosis, while the latter leads with high frequency to inhibition of germ-cell proliferation and germ-line development (Fig. 2e), mimicking the previously described PRC2 null phenotype[38]. Residual PRC2 activity is essential for the survival of oncohistone-containing human cells, illustrating that the K27M-driven tumors are not simply suffering from loss of PRC2 activity, but that tumorigenesis is driven by specific patterns of PRC2 inhibition[23]. In addition to the pattern of oncohistone incorporation, oncohistone levels may also contribute to the PRC2 inhibition and the downstream phenotypes of the mutant cells. We found that introduction of K27M into the germ-line-specific H3.3 gene his-74, which is expressed at lower levels compared with his-72[30], did not result in sterility phenotypes, despite partial redistribution of H3K27me3 (Supplementary Fig. 11). Our data therefore suggest that chromatin changes are not always sufficient to trigger endomitosis, and that the observed phenotypes depend on both oncohistone levels and oncohistone distribution. Systematic and quantitative measurements of oncohistone levels in different human cell lines resulted in the same conclusion that oncohistone levels are important in the context of cellular PRC2 levels[58]. Finally, the fact that more wide-spread, H3-like, distribution of the K27M-containing nucleosomes more often phenocopies complete loss of PRC2 may also explain why H3.1K27M human tumors occur less frequently than H3.3K27M-mediated malignancies.

In *C. elegans*, PRC2 activity is mainly required for germ-line development, and the H3.3K27M mutation results in ectopic germ-cell fates, while somatic tissues appear unaffected. In mammalian systems, regulation of PRC2 activity is essential during stem cell differentiation and neuronal development[64,65] and loss of PRC2 can prevent stem cells from differentiation and support their proliferation[66,67]. Most of the cells in H3.3K27M-driven tumors show characteristics of oligodendrocyte precursor cells, suggesting that the driver mutagenesis occurs in undifferentiated cells[41]. The constraints on PRC2 activity enforced by oncohistone incorporation may prevent the dynamic H3K27me3 changes required during differentiation and activate transcriptional programs that support an undifferentiated state and result in tumorigenesis.

The chromatin reorganization caused by oncohistone incorporation and ectopic PRC2 regulation results in a global misregulation of the transcriptome in mutant cells. We found that mutated *C. elegans* germ cells showed ectopic expression of several hundred genes, which fall into diverse categories such as kinases and ion transporters (Fig. 4). Similarly, transcriptional profiling of tumor-derived tissues and single cells showed that many cancer-related pathways were misregulated[9,41]. To single out key factors that link the chromatin changes and the downstream phenotypes, we took advantage of the powerful *C. elegans* genetics for unbiased screening. Surprisingly, we were able to identify a single point mutation in the *C. elegans* JNK homolog

KGB-1 that significantly rescued the aberrant replication phenotype, making misregulation of the JNK pathway an important contributor to the replicative fate induced by the oncohistone (Fig. 5). The JNK pathway is a known regulator of calcium levels in *C. elegans* gonads and linked to calcium signaling in human cells[45,46,68] and the suppressor mutation indeed rescued the changes in Ca$^{2+}$ levels caused by the H3.3K27M mutation. This suggests that the changes leading to ectopic replication in worms and possibly tumorigenesis in humans are caused by misregulation of specific signal transduction pathways rather than global chromatin changes and the resulting global misregulation of genes. KGB-1/JNK was significantly enriched in the worm RNA-seq analysis and showed significant changes in H3K27me3 occupancy, but it would not have been possible to identify it as a causal mutation based on these criteria alone. JNK itself was not identified as significantly misregulated in single-cell RNA-seq of human K27M tumors. Remarkably, however, the top hit among the differentially expressed genes between H3.3K27M tumor cells and normal brain cells is immediate early response protein 2 (IER2), a suppressor of neuronal differentiation upon JNK-dependent induction[41,69]. Platelet-derived growth factor receptor alpha (PDGFRA), which is often amplified in the K27M-derived tumors, is one of the upstream regulators of JNK[70]. Finally, the causative relationship between aberrant JNK activity and replicative fate has been established in many types of gliomas[49–51]. We found that inhibition of JNK suppresses proliferation of cells derived from K27M tumors more strongly than cells derived from non-K27M gliomas (Fig. 6). This illustrates the utility of genetic models like *C. elegans* for the identification of phenotypic drivers with only moderate expression changes in the mutant cells. We show that inhibition of JNK for a reduction of cell proliferation may be applicable to human tumor cells, thus offering potential for the development of future K27M tumor therapies.

Taken together, our results establish *C. elegans* as a powerful model to understand how chromatin changes result in aberrant cell fates, and identify the JNK pathway as a potential drug target for treatment of H3.3K27M-positive gliomas.

## Methods

***C. elegans* strains and culture**. *C. elegans* strains were grown using standard OP50 feeding conditions for maintenance, stainings and sterility quantifications, and using peptone-rich plates seeded with NA22 for large quantities necessary for ChIP-seq experiments. N2 (Bristol strain) was used as wild type. A list of strains used in this study, including the numbers of alleles generated, is given in Supplementary Table 1. For simplicity, the term H3.3 refers to *his-72*, which is expressed in all cells at all developmental stages both in the soma and in the germ line, throughout the paper[30,34]. All genome edits were generated at the endogenous loci of *his-72* and *kgb-1* using CRISPR/Cas-9 as described in[71]. Briefly, sgRNAs and Cas9 were delivered as plasmids, and repair template as single strand oligonucleotides. To change the chaperone specificity for HIS-72, the H3.3-specific motif AAIG was replaced with the H3-specific motif SAVM. sgRNAs and repair templates, as well as PCR primers used to detect and sequence the mutations, are listed in Supplementary Table 2.

**Sterility quantification**. For sterility levels at 20 °C, 100 L4 larvae were singled, and plates were scored for presence or absence of progeny after 3 days. For sterility levels at 25 °C, L4 larvae were shifted to 25 °C, and 100 of their F1 progeny were singled as L3 larvae (before any germ-line phenotype is visible) and kept at 25 °C. F1 worms with at least one viable F2 offspring were categorized as fertile. To determine the cause of sterility (endomitosis, absence of gonads, under-developed gonad), the 100 single worms were examined by light microscopy. Sterility levels comparisons were performed in at least three biological replicates, and different strains/conditions were compared using student's *t*-test. For sterility in context of RNAi, both F1 mothers and scored F2 progeny were grown on bacteria expressing the desired RNAi constructs. For sterility in context of JNK inhibition, both F1 mothers and scored F2 progeny were grown on OP-50 plates containing 50 μM JNK inhibitor SP600125 (Milipore Sigma, S5567). Boxplots were drawn using default parameters in R, in these and all subsequent quantifications.

**Stainings and imaging**. Worm gonads were dissected in anesthetizing buffer (50 mM sucrose, 75 mM HEPES pH 6.5, 60 mM NaCl, 5 mM KCl, 2 mM MgCl$_2$, 10

mM EGTA pH 7.5, 0.1% NaN$_3$). For antibody stainings, gonads were fixed in methanol for 20 min and acetone for 10 min at −20 °C. Slides were washed three times with PBS for 5 min and incubated with H3K4me3 antibody (Abcam ab8580, 1:10,000), H3K27me2me3 antibody (Active Motif 39535, 1:300), OLLAS antibody (Novus Biologicals NBP1-06713B, 1:300), H3 phosphoS10 antibody (Abcam ab5176, 1:1000), POLD2 antibody (Thermo Fisher Scientific PA5-55401, 1:100), or RAD-51 antibody (Novus Biologicals 29480002, 1:5000) overnight at 4 °C. Slides were washed in PBS three times for 5 min and incubated with a Cy3- or Cy5-conjugated secondary antibodies (Jackson Immunoresearch, 1:700) for 1.5 h at room temperature. Samples were washed with PBS three times for 5 min, stained with DAPI, and mounted with VECTASHIELD Antifade Mounting Medium. Images were obtained using a Leica DM5000 B microscope or a Leica SP8 confocal microscope. Confocal images shown are maximum projections of 0.2–0.4 µm Z-sections of the entire nuclei, except for images showing whole gonads for which 0.8–1 µm Z-sections were taken.

For DAPI intensity quantification, gonads were dissected as describe above, fixed in anesthetizing buffer containing 2% paraformaldehyde (PFA) for 5 min, and permeabilized in PBS containing 0.1% Triton X-100 for 5 min. Gonads were washed in PBS, stained with DAPI in PBS (1:500) and washed once more in PBS. Z-stacks of DAPI-stained gonads were merged with maximum intensity. DAPI signal was measured for wild-type oocytes, intestine nuclei and endomitotic oocytes using the Area Integrated Intensity function in Fiji[72]. DNA content was estimated based on the DAPI measurements and linear regression of values for wild-type oocytes (which are in diakinesis I stage; 4 n) and intestine nuclei (which are endoreduplicated to the DNA content of 32n).

Quantification of different H3.3 proteins, *his-72* and *his-74*, was performed on the live imaging of GFP-tagged version of the proteins from ref. [30] using the Area Integrated Intensity function in Fiji[72].

**BrdU labeling**. Worms were labeled with BrdU on plates as described in ref. [73] with minor modifications. Two milliliters of *E. coli* MG1693 (obtained from the *E. coli* stock center) overnight culture were added to 100 ml of M9 buffer supplemented with 1% glucose, 1.25 µg/ml thiamine, 0.5 µM thymidine, 1 mM MgSO$_4$, and 20 µM BrdU (BD-Pharmingen 550891). Bacteria were grown for 24–36 h at 37 °C in the dark, spun down, resuspended in 1 ml of M9 buffer and used to seed M9 agar plates.

Wild-type and H3.3K27M mutant young adult worms that had been maintained for one generation at 25 °C were put on feeding plates for 4 h at 25 °C. Gonads were dissected as described above and fixed in methanol for 1 h at −20 °C. Slides were blocked in PBSB (PBS containing 0.5% BSA) for 15 min, washed in PBS, fixed with 1% PFA for 15 min and again washed in PBS. To denature DNA and expose the epitope, the slides were incubated with 2 M HCl for 20 min and then neutralized with 0.1 M sodium borate (pH 8.5) for 15 min. Samples were re-blocked in PBSB for 15 min and incubated with anti-BrdU antibody (BD-Pharmingen 555627, 1:200 in PBSB) overnight at 4 °C. The slides were washed three times in PBSB, incubated with secondary antibody (anti-mouse Alexa 488, 1:1000) for 2 h at room temperature, washed again in PBS, counterstained with DAPI and mounted with Vectashield mounting media.

**Western blotting**. Adult worms were washed three times with M9, resuspended in lysis buffer (10 mM Tris pH = 7.5, 100 mM NaCl, 1 mM EDTA, 10% glycerol, 0.5% NP-40) and frozen in liquid nitrogen. Samples were sonicated (10 cycles, 30 s sonication, 30 s rest, snap freezing between each cycle) and spun down (5 min, max speed) to pellet debris. Protein concentrations were measured with the Bio-Rad Protein Assay (Bio-Rad, 5000006). A total of 20 µg (or 2 µg for H3 loading control) was loaded SDS-page gels. Western blotting was performed according to standard procedures using the LI-COR Odyssey system. Primary antibodies (anti-H3K27me2me3, Active Motif 39535, 1:1000; anti-OLLAS, Novus Biologicals NBP1-06713B, 1:1000 or anti-histone H3, Abcam, ab1791, 1:4000) were incubated overnight at 4 °C, and IRDye® secondary antibodies appropriate for each primary antibody were incubated for 1 h at room temperature.

**ChIP-seq**. Native ChIP was performed as described in ref. [74] with minor modifications. Briefly, synchronized worm populations were grown on 15-cm plates seeded with *E. coli* NA22. Worms were harvested as young adults, washed in M9 and resuspended in buffer A (15 mM Tris pH = 7.5, 2 mM MgCl$_2$, 340 mM sucrose, 0.2 mM spermine, 0.5 mM spermidine, 0.5 mM PMSF) supplemented with 0.5% NP-40 and 0.1% Tx-100. Total nuclei, which are enriched for germ-cell nuclei based on morphologic analysis, were obtained by light grinding of the worms under liquid nitrogen followed by douncing, low speed spin and washing of the nuclei. Chromatin was digested to mononucleosomes using MNase (NEB M0247S) at 37 °C using 5 µl of enzyme per 150 µl of nuclei in 1000 µl of the buffer. Each sample was divided into six parts that were incubated for 3, 5, 8, 10, 12, and 15 min, respectively, and then pooled together in order to obtain nucleosomal ladders. Following digestion, chromatin was extracted, solubilized and pre-cleared with empty beads. The soluble chromatin fraction was incubated with primary antibodies against H3K27me2/me3 (Active Motif 39535), OLLAS (Novus Biologicals NBP1-06713B) or H3K36me3 (Abcam ab9050) overnight at 4 °C. Antibody-bound chromatin was captured with magnetic beads for 1 h at 4 °C (50 µl 1:1 mixture of

Dynabeads Protein A and Dynabeads Protein G, Invitrogen). DNA from both input and ChIP samples was then extracted using phenol/chloroform. Libraries were prepared using NEBNext® Ultra™ II DNA Library Prep with sample purification beads using 100 ng of input DNA. Libraries were sequenced using the TruSeq SBS HS v3 chemistry on an Illumina HiSeq 2500 sequencer.

**Oocyte whole-genome sequencing**. Endomitotic oocytes or proximal parts of wild-type gonads were dissected. Samples from 20 to 25 worms were pooled and sonicated using a Covaris instrument, and libraries were prepared using TruSeq DNA-nano kit from Illumina. Whole genomes of the oocytes were sequenced using an Illumina HiSeq 2500. Reads were mapped to *C. elegans* reference genome and normalized to the total number of reads for each sample.

**Sequence analysis**. Paired-end reads (length = 150 bp) were mapped to the *C. elegans* reference genome WBcel215 with Novoalign software (default parameters, producing SAM format files). Reads were transformed into bins of the desired size (JSON format) with a custom UNIX script. JSON files were transformed into bedgraphs annotating maximum amount of reads present in each bin with a custom R script. All files were normalized to the same number of reads. For each ChIP experiment, IP files were normalized using the input file with the corresponding bin size. Biological replicates of ChIP-seq experiments that passed the quality control were merged on the level of aligned reads (SAM files), and the downstream pipeline was performed again using the merged file and smoothed to obtain the final normalized bedgraph for each experiment. Processed data files are bedgraphs containing normalized ChIP results. Normalization was performed by input subtraction (to avoid low number of read biases; marked in the file name by _delta) or by division of input followed by log2 transformation (marked in the file name by _logratio). Bin sizes used in the analysis were 100 bp (marked in the file name by _100bp), 1 kb (marked in the file name by _1kb) or 10 kb (marked in the file name by _10kb). The appropriate bin size was selected for each analysis. Genome browser images were obtained using the UCSC genome browser (100 kb windows) or with Sushi Bioconductor package (whole-genome views). H3K27me3 occupancy domains were called with custom Python and R scripts, heatmaps and hierarchical clustering were performed with Morpheus software (https://software.broadinstitute.org/morpheus).

**Suppressor screen**. The screen for suppressors of H3.3K27M-induced sterility was performed as described in ref. [75]. Briefly, H3.3K27M L4 worms were washed in M9 and incubated in 50 mM ethyl methanesulfonate (EMS) for 4 h at 20 °C. Worms were washed in M9 again and recovered on OP50-seeded NGM plates. Pools of 10 worms each were transferred to fresh NGM plates (100 worms total) and were allowed to lay eggs for 6 h after reaching the adulthood, resulting in ~100 F1 eggs each. Plates were maintained at 25 °C, and F1 adults were washed off at the second day of adulthood to retain staged F2 eggs. F2 worms were maintained at 25 °C, and 2000 of non-sterile looking worms were singled. Plates that contained progeny after three more generations at 25 °C were retained for further analysis. Mapping of the suppressor mutation was performed as described in ref. [76]. Briefly, suppressor strains were backcrossed to the original H3.3K27M strain, and 30 independent F2 plates homozygous for the suppressor mutation (scored based on fertility levels at 25 °C) were pooled together for total DNA isolation. Samples were sonicated using a Covaris instrument, and the libraries were prepared using TruSeq DNA-nano kit from Illumina. Whole genomes from the suppressor strains as well as from original H3.3K27M mutant worms were sequenced using an Illumina HiSeq 2500. Reads were mapped to *C. elegans* reference genome, and SNPs were identified using the High-Throughput Sequencing portal of the EPFL Bioinformatics and Biostatistics core facility (hts station). Candidate causative mutations were identified as SNPs with nearly 100% presence in suppressor strains when compared with the H3.3K27M mutant strain. Presence of the mutation was confirmed by Sanger sequencing. Suppressor function of the mutation was confirmed by introducing the same mutation into H3.3K27M worms using CRISPR/Cas9 genome editing.

**Calcium level detection**. Measuring of gonadal calcium levels was carried out as described in ref. [77]. Briefly, adult worms were injected with 100 µM Calcium Green-1 dextran, 10,000 MW (Molecular Probes). Worms were rested for 2 h and imaged with a fluorescence microscope. Levels of Ca$^{2+}$ were estimated by measuring intensity of green fluorescence in multiple equally sized areas in each gonad using ImageJ.

**KGB-1 alignment/modeling**. The KGB-1 amino acid sequence was obtained from WormBase and entered into the SWISS-MODEL structure prediction software[78]. The software identified and used human JNK1 crystal structure (PBD ID 3VUG) as a template for the structure prediction. GMQE (Global Model Quality Estimation) and QMEAN Z-score for the obtained model equal 0.74 and −3.83, respectively, indicating that the model is of good quality and high fidelity. To determine if S287 is a potential phosphorylation site we utilized serine, threonine or tyrosine phosphorylation site prediction software NetPhos3.1[79].

**Table 1 Values of coefficients in the formula approximating the oncohistone and H3K27me3 dependency**

| | |
|---|---|
| A0 | −0.709110206 |
| A1 | 0.689422042 |
| A2 | 0.92135143 |
| A3 | 0.291658614 |
| A4 | −0.153205021 |
| A5 | −0.049106574 |
| A6 | 0.002409524 |
| A7 | 0.00175899 |
| A8 | 0.000128287 |

**Function approximation**. To model the dependency between oncohistone incorporation and H3K27me3 levels, a formula approximating the 1 kb logratio ChIP-seq data from wild-type and H3.3K27M mutant worms was identified, where $x$ = pre-existing (wild-type) levels of H3K27me3, $y$ = H3.3K27M levels of H3K27me3, and $z$ = resulting (observed in H3.3K37M mutant) H3K27me3 levels. The approximation was performed in 3-steps. First, the formula for obtaining $z$ values based on the experimental $x$ and $y$ was proposed for the first 20 data points. Then that formula was applied to all of the data points and best-fit function was approximated using Solver, a Microsoft Excel add-in program. The error was optimized by maximizing the $R^2$ correlation value and minimizing the formula error understood as $\sum_{i=1}^{100281}(Z_i - z_i)^2$, where 100281 is the number of bins in the genomic data, $Z_i$ are the $z$ values obtained by the function, and $z_i$ are the experimental $z$ values. That procedure was repeated in several iterations with different base functions, and the use of an approximation based on the Taylor's theorem resulted in the best fit:

$$Z = A_0 + A_1 \times \sum_{n=1}^{7} A_{n+1}(x-y)^n$$

with the A values listed in Table 1.

**Human cell culture**. The SF8628 cell line was purchased from Merck (SCC127), and SF9402 and SF9427 cell lines were kindly provided by Rintaro Hashizume (Northwestern University). The proliferation assays in presence of DMSO or increasing concentration of the JNK inhibitor SP600125 were performed as described in ref. [52]. Briefly, 5000 cells per well were plated in 96-well plates (in duplicates) and incubated in presence of the drug for 72 h. Relative numbers of viable cells in each well were determined using CellTiter 96 AQueous One Solution Cell Proliferation Assay (Promega). Experiments were performed in six biological replicates, and differences between cell lines at each drug concentration were calculated using one-way ANOVA.

**RNA-seq**. RNA was isolated from dissected gonads of N2 and H3.3K27M mutant worms grown at 25 °C using Trizol, in biological replicates for both samples. RNA quantification, library preparation and data analysis were performed as described in ref. [30]. Briefly, the TruSeq mRNA stranded kit from Illumina was used for the library preparation with < 500 ng of total RNA as input. Reads of 50 bases were generated using the TruSeq SBS HS v3 chemistry on an Illumina HiSeq 2500 sequencer. The reads (length = 50 bp) were mapped with the TopHat v2.0.13 (default parameters) software to the *C. elegans* reference genome (WBcel235). The normalization and differential expression analysis between N2 and H3.3K27M mutant samples were performed with the R/Bioconductor package edgeR v.3.10.5, for the genes annotated in the reference genome. Genes with a *p*-value with FDR < 0.05 and fold change > 2 were considered significant. Raw and normalized counts as well as statistical measures are provided in Supplementary Data 1.

**Reporting summary**. Further information on research design is available in the Nature Research Reporting Summary linked to this article.

## Data availability

All relevant data supporting the key findings of this study are available within the article and its Supplementary Information files or from the corresponding author upon reasonable request. Strains, reagents, and scripts are available upon request. Sequencing data has been deposited at the Gene Expression Omnibus (GEO) under accession number GSE117533. The source data underlying Figs. 1a,c,d, 2c–e, 5b,c,e,g,h, 6b and Supplementary Figs 1 and 11b are provided as a Source Data files. A reporting summary for this Article is available as a Supplementary Information file.

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

## Acknowledgements

We thank Nicolas Roggli for help with the mapping and analysis pipeline for the sequencing data. We received cell lines from Rintaro Hashizume. The RAD-51 antibody was a kind gift from Susan Gasser. Mateusz Mendel, Sotiris Sotiriou, Ramesh Pillai, and Thanos Halazonetis provided reagents and help with the mammalian cell cultures. Maksym Shyian provided help with the suppressor screen analysis. Maciej Dorobek provided help with mathematical analysis of the genomic data. Thanos Halazonetis, Robbie Loewith, Ramesh Pillai, David Shore, and Monica Gotta provided comments on the paper. The Bioimaging Center of the Faculty of Sciences and the iGE3 Genomics Platform at the University of Geneva provided imaging and sequencing services as well as data analysis. Some C. elegans strains were provided by the CGC, which is funded by NIH Office of Research Infrastructure Programs (P40 OD010440). The work was funded by the Swiss National Science Foundation (Grants #31003A_156774 and #31003A_156774), the Republic and Canton of Geneva, and an iGE3 PhD Students Award to K.D.

## Author contributions

F.A.S. and K.D. conceived the study, K.D., J.M.W. and F.A.S. carried out experiments, K.D., M.S. and A.P. analyzed the data, K.D. and F.A.S. prepared the paper, and F.A.S. acquired funding.

## Additional information

**Competing interests:** The authors declare no competing interests.

