## [Peer Review File · Nature Communications]

Reviewers' comments:

Reviewer #1 (Remarks to the Author):

This study describes the introduction of the oncogenic H3.3 mutation resulting in lysine-to-methionine substitution at residue 27 (H3.3-K27M) of the *his-72* locus in *C. elegans*. While this particular mutation was discovered in human cancer (pedGBM), and detailed analysis has been performed as far as studies on the tumor biopsy could be performed, large parts of the mechanism behind the mutation remains poorly investigated (even from recent high-profile papers on K27M). This study therefore provides a novel insight into what happens when K27M becomes incorporated into H3.3 deficient sites, guided by the canonical H3 as a vehicle for ectopic placements. It has been reported that K27M blocks PRC2 activity and that it leads to both loss and redistribution of H3K27me3. The authors report complete sterility from underdeveloped gonads in worms with K27M. The strong influence on PRC2 activity by K27M encouraged the authors to find the evidence for sterility from aberrant patterns of H3K27me3 depositions. This was supported by a genetic suppressor screen of K27M worms identifying *kgb-1*, a JNK1 homologue, indicating downstream signaling effect from remodeled H3K27me3 patterns.

Their analysis of K27M when residing on H3.1 compared to H3.3 provides additional insight into deposition and function of H3K27me3. The statement "pre-existing H3K27me3 can locally sufficiently stimulate PRC2 to overcome the inhibitory effects of oncohistone incorporation" epitomize this notion.

Major comments

Page 11, 17 K27M in H3.3 results in grandeur loss globally and surprising gain of H3K27me3 at some loci in pedGBM. There is a discrepancy between loss and gain and maintenance of high levels of H3K27me3 that cannot be explained by H3-like K27M in worms, since those only represent ectopic incorporation of K27M. It would be great if the authors could develop this discussion a bit further. In addition, K27M when incorporated cannot be methylated due to the Met, but neighboring K27s can and will be when surrounded by high H3K27me3 levels in particular. This indicate two independent operating mechanisms binding different parts of PRC2, the first one SUZ12/EZH2 and the other one EED. This would go in line with crystal structures analysis recently published by others.

Page 15 The suppressor screen is a great experiment the authors conducted, but it leaves me wonder why they did not talk about additional mutant alleles than the *kgb-1* (S287N) likely produced. Did the authors really only get one allele? I think it would be nice if they could discuss other allelic families they got from the screen as that would strengthen the argument of a downstream signaling cascade and perhaps other pathways.

Minor comments

Row 169 "H3.3 to and H3-like" should be "H3.3 to an H3-like".

Row 186 It would be interesting to see the incorporation over genic regions, e.g. 5'UTR-TSS-Body-TTS-3'UTR.

Row 270 The referral to Fig. 4c should be 4b.

Row 295 The data in Fig. 5d better be reproduced with the mut JNK/KGB-1 S287N ChIP-seq data of H3K27me3, if the authors have it.

Row 408 Recall that a small percentage of H3.1 also is mutated in pedGBM, which suggest that one can not categorically separate H3.3 and H3.1 based on the "replicative fate and endomitosis" and "inhibition of germ cell development". How does that reconcile with the fact that both generate DIPGs?

Figure comments

2a/b It has not been explained what the star (*) indicates. If it is the X-chromosome, how can the authors be sure if it has not been stained for (rather than H3K27me3 staining)?

2e Considering the clear evidence for "endomitosis" and "no gonads" in mut, isn't the %

“underdeveloped” gonads surprisingly low?

Reviewer #2 (Remarks to the Author):

This manuscript describes an investigation into the phenotypic consequences of the presence of a mutation in a core histone variant, H3.3, that prevents methylation of lysine 27. This mutation has been associated with formation of certain tumors. Although the mutation has been studied in certain systems, it has not been directly linked to changes in proliferation previously. Here, the authors create the mutation in *C. elegans* and identify a germ cell-specific effect on proliferation. They demonstrate that the mutation can directly interrupt normal cell differentiation. They next examine the molecular effects on H3K27me3 using immunofluorescence and ChIP-seq experiments, and show that the mutant blocks H3K27me3 in regions of the genome where it was not particularly high (e.g. many places on autosomes), but that in other regions where H3K27me3 was abundant (e.g. the X chromosome), the mutant did not disrupt the pattern of modification. A suppressor screen in *C. elegans* identified a component of JNK signaling pathway as a key mediator of the proliferation defects, and demonstrated that inhibiting JNK signaling in mammalian cells bearing the H3.3K27M mutation was more potent in disrupting proliferation than in wild type cells.

I'm not entirely sure how much more it adds to the H3K27M story as it pertains to the oncohistone aspect, but as far as the new insight regarding *C. elegans* germ cell biology, it has some interesting contributions. Histone modifications are clearly essential in establishing proper gene expression patterns in the germ line and H3K27me3 is a critical component. So the observation that this mechanism can exist and influence the deposition and accumulation of this mark will be important for future studies.

Major:

The authors attempted to create this mutation in two different H3.3 genes, but only one gave the phenotype. The authors argued that it was due to expression level differences, but there is no data to back this up. What if some of the H3.3 genes have cell type specificity? Perhaps this successful case is not the norm but the product of some unusual circumstances in the germ line that are not relevant to the broader effect.

The K27me3/K36me3 dynamic has an unusual mechanism in the germ line, especially with regard to the X vs A expression relationship, and perhaps this dynamic is critical to the phenotype. The authors do not look at the effects on H3K36me3 in this background, which would provide important context. Do H3K36me3 domains expand when the H3K27me3 domains contract?

The authors briefly indicate that H3.3K27M is preferentially associated with open chromatin from other studies and proffer the evidence that it is excluded from the silenced X chromosome in germ cells as confirmation. They say little however about its distribution on autosomes. There are clear expectations about specific genes on the autosomes that should have open chromatin, but no discussion of whether H3.3K27M associates with those genes. Is this true? Similarly, was there any preference for the genes that changed expression in the mutant conditions to be known to have germline expression (or not)? Figure 4d is not particularly informative; it would be more useful to know where these genes were expressed or at least whether they fell into germline or soma-enriched categories.

Minor grammatical notes:

The plural of progeny is progeny, not progenies (eg Fig 2e, Fig 5a).

line 169 typo – an, not and

Response to reviewers' comments:

Delaney et al.
NCOMMS-19-00234

We thank both reviewers for their helpful comments and feedback. We have addressed all their comments and suggestions, as detailed below. Based on community feedback on our preprint version of the manuscript, we would also like to simplify the title of the manuscript to "H3.3K27M-induced chromatin changes drive ectopic replication through misregulation of the JNK pathway."

Reviewer #1 (Remarks to the Author):

This study describes the introduction of the oncogenic H3.3 mutation resulting in lysine-to-methionine substitution at residue 27 (H3.3-K27M) of the his-72 locus in *C. elegans*. While this particular mutation was discovered in human cancer (pedGBM), and detailed analysis has been performed as far as studies on the tumor biopsy could be performed, large parts of the mechanism behind the mutation remains poorly investigated (even from recent high-profile papers on K27M). This study therefore provides a novel insight into what happens when K27M becomes incorporated into H3.3 deficient sites, guided by the canonical H3 as a vehicle for ectopic placements. It has been reported that K27M blocks PRC2 activity and that it leads to both loss and redistribution of H3K27me3. The authors report complete sterility from underdeveloped gonads in worms with K27M. The strong influence on PRC2 activity by K27M encouraged the authors to find the evidence for sterility from aberrant patterns of H3K27me3 depositions. This was supported by a genetic suppressor screen of K27M worms identifying *kgb-1*, a JNK1 homologue, indicating downstream signaling effect from remodeled H3K27me3 patterns.

Their analysis of K27M when residing on H3.1 compared to H3.3 provides additional insight into deposition and function of H3K27me3. The statement "pre-existing H3K27me3 can locally sufficiently stimulate PRC2 to overcome the inhibitory effects of oncohistone incorporation" epitomize this notion.

Major comments

Reviewer 1 Comment #1

Page 11, 17 K27M in H3.3 results in grandeur loss globally and surprising gain of H3K27me3 at some loci in pedGBM. There is a discrepancy between loss and gain and maintenance of high levels of H3K27me3 that cannot be explained by H3-like K27M in worms, since those only represent ectopic incorporation of K27M. It would be great if the authors could develop this discussion a bit further. In addition, K27M when incorporated cannot be methylated due to the Met, but neighboring K27s can and will be when surrounded by high H3K27me3 levels in particular. This indicate two independent operating mechanisms binding different parts of PRC2, the first one SUZ12/EZH2 and the other one EED. This would go in line with crystal structures analysis recently published by others. *Our data, as well as the data from pedGBM, and from structural and functional studies of PRC2 support a model where PRC2 is allosterically activated by nucleosomes containing H3K27me3 to establish methylation marks on the neighboring nucleosomes, and that*

incorporation of H3.3K27M inhibits this nucleosome-to-nucleosome spreading. This model takes into consideration that only a minority of nucleosomes contain H3.3K27M, but these are sufficient to interfere with the allosteric activation of PRC2 and inhibit the spreading of H3K27me3 domains. This likely results in a further reduction of PRC2 recruitment, until the heterochromatin domain is lost. The gain in H3K27me3 levels is indeed harder to explain from our worm data, but we speculate that with the loss of H3K27me3 domains over large parts of the genome more PRC2 becomes available and can be recruited by domains where incorporation of H3.3K27M is low. This is supported by the observation of MES-2/PRC2 accumulation in the mut worms in Figure 2 a-b. We have added a paragraph discussing this in more detail on page 18 lines 386-407, and have included a model figure (Fig. 7) that illustrates these dynamics. As the reviewer points out, the model is indeed consistent with recent structural data of PRC2, showing that both EZH2 and EED are important for efficient methyltransferase activity. However, it is difficult to directly extend the findings of these studies to our worm model.

Reviewer 1 Comment 2#

Page 15 The suppressor screen is a great experiment the authors conducted, but it leaves me wonder why they did not talk about additional mutant alleles than the *kgb-1* (S287N) likely produced. Did the authors really only get one allele? I think it would be nice if they could discuss other allelic families they got from the screen as that would strengthen the argument of a downstream signaling cascade and perhaps other pathways.

*The easiest way to suppress the sterility phenotype is by mutation of the *his-72* gene that cause loss of the HIS-72 protein, but we were not interested in these alleles. In addition to the *kgb-1* alleles that we characterized, we indeed found additional alleles that suppress the phenotype to a lesser degree. For these alleles, it was however difficult to identify the causal mutations using the approach we took, as it relies on a backcross with the original strain and the subsequent identification of plates homozygous for the suppressor mutation. The unambiguous identification of homozygous plates proved challenging for the weak suppressor alleles, and for this reason, we did not pursue them further. We agree that the suppressor screen was likely not saturated, and that additional pathways could be identified by repeating it. We have added a sentence in the text commenting on additional alleles (page 14 line 297-299).*

Minor comments

Reviewer 1 Comment #3

Row 169 “H3.3 to and H3-like” should be “H3.3 to an H3-like”.

The change has been implemented.

Reviewer 1 Comment #4

Row 186 It would be interesting to see the incorporation over genic regions, e.g. 5'UTR-TSS-Body-TTS-3'UTR.

A similar comment was brought up by Reviewer 2 (#3). We added average plots illustrating H3.3 incorporation over and around genes, in quintiles of gene expression level (Fig. 4 d-e). We also repeated this analysis for H3K27me3 levels, for both wild-type and mut worms (Fig. 4 panels d-e). These plots confirm previous findings that H3.3 incorporation correlates with gene expression levels, while H3K27me3 levels are anti-correlated with gene expression levels. They also show that the patterns of H3.3 around genes remain largely unchanged

upon mutation of H3.3K27M, but that H3K27me3 levels are affected. Specifically, we find that genes with expression changes in the mutant fall mostly into wild type expression quintiles 2 and 3, which show both H3.3 incorporation and at least some level of H3K27me3 in wild type worms (Fig. 4f). Accordingly, we added a description to the text on page 13 lines 279-288.

Reviewer 1 Comment #5

Row 270 The referral to Fig. 4c should be 4b.

The change has been implemented.

Reviewer 1 Comment #6

Row 295 The data in Fig. 5d better be reproduced with the mut JNK/KGB-1 S287N ChIP-seq data of H3K27me3, if the authors have it.

We have added the track showing H3K27me3 data from the mut JNK/KGB-1 S287N to Fig. 5d. It shows that the JNK/KGB-1 S287N suppressor mutation does not affect the loss of H3K27me3 over the kgb-1 gene, and that the suppression of the sterility phenotype likely occurs downstream of the chromatin defects. To further illustrate this point, we have added an additional supplemental figure (Supplementary Fig. 9) showing the genomic H3K27me3 distribution in mut JNK/KGB-1 S287N worms compared to wild type and mut animals on parts of chromosomes II and X, as well as genome wide correlation plots.

Reviewer 1 Comment #7

Row 408 Recall that a small percentage of H3.1 also is mutated in pedGBM, which suggest that one can not categorically separate H3.3 and H3.1 based on the “replicative fate and endomitosis” and “inhibition of germ cell development”. How does that reconcile with the fact that both generate DIPGs?

This is a good point. We did not intend to imply that mutation of H3.1 always leads to a different phenotype than mutation of H3.3. Indeed, we observe the same spectrum of phenotypes in both the H3.3 and the H3-like H27M mutants, and it is the frequencies of specific phenotypes that differ. We propose that the distribution of oncohistone incorporation (and likely also the levels; see response to Reviewer 2 comment #1) can have a strong influence on the H3K27me3 distribution and thus the phenotype. We observe that in our worm model, a redistribution of the oncohistone over larger parts of the genome results in a stronger loss of H3K27me3, and that the phenotype shifts from endomitosis towards sterility. This provides an explanation for why the K27M mutation is more prevalent in H3.3 than in H3.1, but it does not exclude the possibility that mutation of H3.1 also leads to replicative fates, as we observe a small percentage of H3-like mut worms with endomitotic oocytes. We have clarified this in the discussion on pages 20-21 lines 442-449.

Figure comments

Reviewer 1 Comment #8

2a/b It has not been explained what the star (*) indicates. If it is the X-chromosome, how can the authors be sure if it has not been stained for (rather than H3K27me3 staining)?

We added the asterisk description to the legend of Fig. 2: “Chromosome X was identified by depletion of H3.3 and H3K4me3 staining shown in Fig. S3, and is marked with an asterisk”. In the wild type and mut worms, we identified chromosome X by depletion of H3.3 and H3K4me3. For the H3-like mut worms, we relied only on the latter, and we have added the

corresponding panels to Fig. S3. The text still states that the chromosome X is identified by co-staining with H3K4me3. Page 7, lines 142-145.

Depletion of H3K4me3 staining has previously been established as a marker for chromosome X (Kelly et al. 2002, Delaney et al. 2018), and H3K4me3 patterns do not change upon K27M mutation. We chose to show the H3K4me3 staining as a supplemental figure (Supplementary Fig 3) to avoid overcrowding of Fig. 2.

Importantly, the depletion of H3.3 and the enrichment of H3K27me3 on the X chromosome are also clearly evident from our genomic data in Fig. 3, and we feel that together, our data provides sufficient support of the identification of the X chromosome in the staining. We comment on this on page 9, lines 185-189 and page 10, lines 199-202.

Reviewer 1 Comment #9

2e Considering the clear evidence for “endomitosis” and “no gonads” in mut, isn't the % “underdeveloped” gonads surprisingly low?

Based on the inhibition of PRC2 by the K27M mutation, we expected a “no gonad” phenotype, as this is also the phenotype of PRC2 loss-of-function mutants in C. elegans. We were surprised to find that instead the endomitosis phenotype was prevalent in H3.3K27M mutant worms, as described in the manuscript. Additionally, we observed underdeveloped gonads at low frequency, and because these were clearly distinguishable from the “absence of gonad” phenotype, we felt a separate category was warranted. We do not fully understand what causes the underdeveloped gonad phenotype, but we speculate that this may occur when germ cells lose their identity. We base this on the observation that the germ cell nuclei in these worms have a different morphology from pachytene nuclei. However, due to the low frequency of this phenotype, it was difficult to thoroughly analyze it.

Reviewer #2 (Remarks to the Author):

This manuscript describes an investigation into the phenotypic consequences of the presence of a mutation in a core histone variant, H3.3, that prevents methylation of lysine 27. This mutation has been associated with formation of certain tumors. Although the mutation has been studied in certain systems, it has not been directly linked to changes in proliferation previously. Here, the authors create the mutation in *C. elegans* and identify a germ cell-specific effect on proliferation. They demonstrate that the mutation can directly interrupt normal cell differentiation. They next examine the molecular effects on H3K27me3 using immunofluorescence and ChIP-seq experiments, and show that the mutant blocks H3K27me3 in regions of the genome where it was not particularly high (e.g. many places on autosomes), but that in other regions where H3K27me3 was abundant (e.g. the X chromosome), the mutant did not disrupt the pattern of modification. A suppressor screen in *C. elegans* identified a component of JNK signaling pathway as a key mediator of the proliferation defects, and demonstrated that inhibiting JNK signaling in mammalian cells bearing the H3.3K27M mutation was more potent in disrupting proliferation than in wild type cells.

I'm not entirely sure how much more it adds to the H3K27M story as it pertains to the oncohistone aspect, but as far as the new insight regarding *C. elegans* germ cell biology, it has some interesting contributions. Histone modifications are clearly essential in establishing proper gene expression patterns in the germ line and H3K27me3 is a critical component. So the observation that this mechanism can exist and influence the deposition and accumulation of this mark will be important for future studies.

Major:

Reviewer 2 Comment #1

The authors attempted to create this mutation in two different H3.3 genes, but only one gave the phenotype. The authors argued that it was due to expression level differences, but there is no data to back this up. What if some of the H3.3 genes have cell type specificity? Perhaps this successful case is not the norm but the product of some unusual circumstances in the germ line that are not relevant to the broader effect.

We have previously shown that some H3.3 genes in C. elegans have indeed cell type specific expression patterns (Delaney et al. 2018). For this reason, we introduced the K27M mutation to the his-72 gene, which is expressed ubiquitously in every cell in the worm. We found that the mutation mainly affects the germ line, which may be surprising given the expression pattern of his-72, but is consistent with the previous findings that PRC2 loss-of-function alleles cause mainly germline phenotypes, and that the germline is more susceptible to changes in cell fate than other tissues.

With the germ line phenotype in mind, we also introduced the K27M mutation to his-74, which is expressed in all germ cells, but absent from soma. The HIS-74 levels are significantly lower than the HIS-72 levels in germ cell nuclei, as measured by endogenously attached fluorescent tag (Delaney et al. 2018, and now also quantified in Supplementary Fig. 11). Even though we observe changes in the H3K27me3 patterns in the HIS-74K27M mutant (Supplementary Fig. 11), the worms do not develop sterility phenotypes. We therefore concluded that the lower levels of his-74 expression are insufficient to trigger the phenotype. We have clarified this in the text on page 20 lines 439-444.

Reviewer 2 Comment #2

The K27me3/K36me3 dynamic has an unusual mechanism in the germ line, especially with regard to the X vs A expression relationship, and perhaps this dynamic is critical to the phenotype. The authors do not look at the effects on H3K36me3 in this background, which would provide important context. Do H3K36me3 domains expand when the H3K27me3 domains contract?

This is a good point that we should have discussed. Surprisingly, we find that the H3K36me3 distribution remains largely unchanged in the mut worms, and the H3K36me3 domains neither expand nor shrink. This is consistent with the finding that H3K36me3 domains set boundaries for H3K27m3 domains, and not vice versa, in C. elegans (Gaydos et al. 2012). We have now added Supplementary Fig. 5 showing genome browser views, correlation plots and chromosome-wide levels of H3K36me3 distribution in wild type and mut worms. We also discuss these findings in the text on page 9, lines 189-194.

Reviewer 2 Comment #3

The authors briefly indicate that H3.3K27M is preferentially associated with open chromatin from other studies and proffer the evidence that it is excluded from the silenced X chromosome in germ cells as confirmation. They say little however about its distribution on autosomes. There are clear expectations about specific genes on the autosomes that should have open chromatin, but no discussion of whether H3.2K27M associates with those genes. Is this true? Similarly, was there any preference for the genes that changed expression in the mutant conditions to be known to have germline expression (or not)? Figure 4d is not particularly informative; it would be more useful to know where these genes were expressed or at least whether they fell into germline or soma-enriched categories.

A similar comment was raised by Reviewer 1 (#4). We now show H3.3 and H3K27me3 data as average plots at genes, in quintiles based on gene expression in wild type, in Fig. 4d-e. These plots show that H3.3 associates with highly expressed genes (which are mostly autosomal) in both wild type and mut worms. Genes that change expression are mostly associated with quintiles 2 and 3, where both H3.3 and H3K27me3 are present (new Fig. 4f). Moreover, we now show that there is almost no overlap of the genes with changed expression in the mutant and previously annotated germline genes (Wang et al. 2009) (new Fig. 4g). We discuss this in the text page 13 lines 279-290.

Minor grammatical notes:

Reviewer 2 Comment #4

The plural of progeny is progeny, not progenies (eg Fig 2e, Fig 5a).
We have corrected this in the figures and throughout the manuscript.

Reviewer 2 Comment #5

line 169 typo – an, not and
The change has been implemented.

REVIEWERS' COMMENTS:

Reviewer #2 (Remarks to the Author):

The authors have done a thorough job of addressing the concerns of both reviewers. I am satisfied with the changes and think it is a good paper with thorough and thoughtful analysis.

Response to reviewers' comments:

Delaney et al.
NCOMMS-19-00234A

Reviewer #1 (Remarks to the Author):

None

Reviewer #2 (Remarks to the Author):

The authors have done a thorough job of addressing the concerns of both reviewers. I am satisfied with the changes and think it is a good paper with thorough and thoughtful analysis.

We thank both reviewers for their encouraging feedback.